

# Ecosystem functional response across precipitation extremes in a sagebrush steppe

Andrew T. Tredennick[1,*], Andrew R. Kleinhesselink[1,2,*], J. Bret Taylor[3] and Peter B. Adler[1]

[1] Department of Wildland Resources and the Ecology Center, Utah State University, Logan, UT, United States of America
[2] Department of Ecology and Evolutionary Biology, University of California, Los Angeles, CA, United States of America
[3] United States Department of Agriculture, Agricultural Research Service, U.S. Sheep Experiment Station, Dubois, ID, United States of America
[*] These authors contributed equally to this work.

Corresponding author
Andrew T. Tredennick,
atredenn@gmail.com

## ABSTRACT

**Background**. Precipitation is predicted to become more variable in the western United States, meaning years of above and below average precipitation will become more common. Periods of extreme precipitation are major drivers of interannual variability in ecosystem functioning in water limited communities, but how ecosystems respond to these extremes over the long-term may shift with precipitation means and variances. Long-term changes in ecosystem functional response could reflect compensatory changes in species composition or species reaching physiological thresholds at extreme precipitation levels.

**Methods**. We conducted a five year precipitation manipulation experiment in a sagebrush steppe ecosystem in Idaho, United States. We used drought and irrigation treatments (approximately 50% decrease/increase) to investigate whether ecosystem functional response remains consistent under sustained high or low precipitation. We recorded data on aboveground net primary productivity (ANPP), species abundance, and soil moisture. We fit a generalized linear mixed effects model to determine if the relationship between ANPP and soil moisture differed among treatments. We used nonmetric multidimensional scaling to quantify community composition over the five years.

**Results**. Ecosystem functional response, defined as the relationship between soil moisture and ANPP, was similar among irrigation and control treatments, but the drought treatment had a greater slope than the control treatment. However, all estimates for the effect of soil moisture on ANPP overlapped zero, indicating the relationship is weak and uncertain regardless of treatment. There was also large spatial variation in ANPP within-years, which contributes to the uncertainty of the soil moisture effect. Plant community composition was remarkably stable over the course of the experiment and did not differ among treatments.

**Discussion**. Despite some evidence that ecosystem functional response became more sensitive under sustained drought conditions, the response of ANPP to soil moisture was consistently weak and community composition was stable. The similarity of ecosystem functional responses across treatments was not related to compensatory shifts at the plant community level, but instead may reflect the insensitivity of the

dominant species to soil moisture. These species may be successful precisely because they have evolved life history strategies that buffer them against precipitation variability.

## INTRODUCTION

Water availability is a major driver of annual net primary productivity (ANPP) in grassland ecosystems (*Huxman et al., 2004*; *Hsu, Powell & Adler, 2012*). Therefore, projected changes in precipitation regimes, associated with global climate change, will impact grassland ecosystem functioning. At any given site, the relationship between ANPP and water availability (e.g., soil moisture) can be characterized by regressing historical observations of ANPP on observations of soil moisture. Fitted functional responses can then be used to infer how ANPP may change under future precipitation regimes (e.g., *Hsu, Powell & Adler, 2012*). A problem with this approach is that it requires extrapolation if future precipitation falls outside the historical range of variability (*Smith, 2011*; *Peters et al., 2012*). For example, the soil moisture-ANPP relationship may be linear within the historical range of interannual variation, but could saturate at higher levels of soil moisture. Saturating relationships are actually common (*Hsu, Powell & Adler, 2012*; *Gherardi & Sala, 2015b*), perhaps because other resources, like nitrogen, become more limiting in wet years than dry years. Failure to accurately estimate the curvature of the soil moisture-ANPP relationship will lead to over- or underprediction of ANPP under extreme precipitation (*Peters et al., 2012*).

Another problem with relying on historical ecosystem functional responses to predict impacts of altered precipitation regimes is that these relationships themselves might shift over the long-term. Shifts in species identities and/or abundances can alter an ecosystem's functional response to water availability because different species have different physiological thresholds. *Smith, Knapp & Collins (2009)* introduced the 'Hierarchical Response Framework' for understanding the interplay of community composition and ecosystem functioning in response to long-term shifts in resources. In the near term, ecosystem functioning such as ANPP will reflect the physiological responses of individual species to the manipulated resource level. For example, ANPP may initially decline under simulated drought because the community consisted of drought-intolerant species (*Hoover, Knapp & Smith, 2014*), but functioning may recover over longer time spans as new species colonize or resident species reorder in relative abundance. It is also possible that ecosystem functioning eventually shifts to a new mean state, reflecting the suite of species in the new community (*Knapp, Briggs & Smith, 2012*).

Experimental manipulations of limiting resources, like soil moisture, offer a way to test how ecosystems will respond to resource levels that fall outside the historical range of variability (*Avolio et al., 2015*; *Gherardi & Sala, 2015a*; *Knapp et al., 2017*). Altering the amount of precipitation over many years should provide insight into the time scales at which water-limited ecosystems respond to chronic resource alteration. We propose

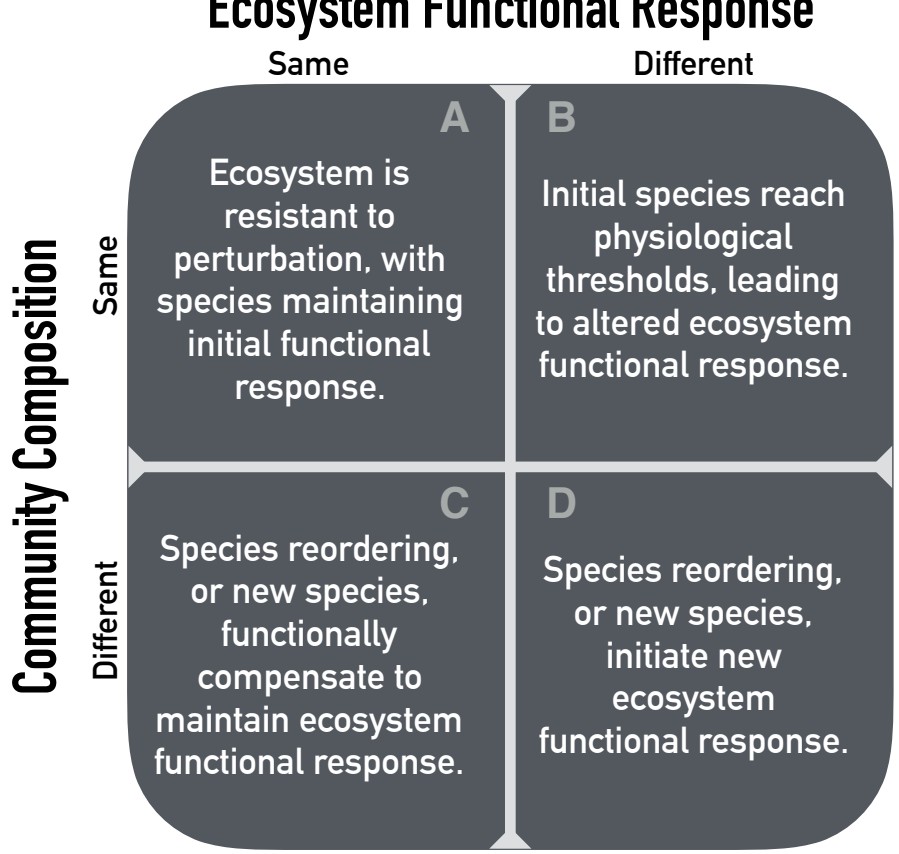

## Ecosystem Functional Response

|  | Same | Different |
|---|---|---|
| **Same** | **A** Ecosystem is resistant to perturbation, with species maintaining initial functional response. | **B** Initial species reach physiological thresholds, leading to altered ecosystem functional response. |
| **Different** | **C** Species reordering, or new species, functionally compensate to maintain ecosystem functional response. | **D** Species reordering, or new species, initiate new ecosystem functional response. |

**Community Composition**

**Figure 1** Possible outcomes of chronic resource alteration based on the 'Hierarchical Response Framework' (*Smith, Knapp & Collins, 2009*).

four alternative scenarios for the effect of precipitation manipulation on the ecosystem functional response to soil moisture based on the Hierarchical Response Framework (Fig. 1). We define 'ecosystem functional response' as the relationship between available soil moisture and ANPP. We focus on soil moisture rather than precipitation because soil moisture is more directly related to plant resource requirements. The four scenarios are based on possible outcomes at the community (e.g., community composition) and ecosystem (e.g., soil moisture-ANPP regression) levels.

First, altered precipitation might have no effect on either ecosystem functional response or community composition (Fig. 1A). In this case, changes in ANPP would be well predicted by the current, observed soil moisture-ANPP relationship. This corresponds to the early phases of the Hierarchical Response Framework, where ecosystem response follows the physiological responses of individual species. Second, the ecosystem functional response might change while community composition remains the same (Fig. 1B). A saturating soil moisture-ANPP response fits this scenario, where individual species hit physiological thresholds or are limited by some other resource. Third, the ecosystem functional response might be constant but community composition changes (Fig. 1C). In this case, changes
in species' identities and/or abundances occur in response to altered precipitation levels and species more suited to the new conditions compensate for reduced function of initial residents. Fourth, and last, both ecosystem functional response and community composition could change (Fig. 1D). New species, or newly abundant species, with different physiological responses completely reshape the ecosystem functional response.

All four outcomes are possible in any given ecosystem, but the time scales at which the different scenarios play out likely differ (*Smith, Knapp & Collins, 2009*; *Wilcox et al., 2016*; *Knapp et al., 2017*). To determine these time scales, we need to amass information on how quickly ecosystem functional responses change in different ecosystems. We also need to understand whether changes at the ecosystem level are driven by community level changes or individual level responses.

To that end, here we report the results of a five-year precipitation manipulation experiment in a sagebrush steppe. We imposed drought and irrigation treatments (approximately ±50%) and measured ecosystem (ANPP) and community (species composition) responses. We focus on how the drought and irrigation treatments affect the relationship between interannual variation in available soil moisture and interannual variation in ANPP, and if community dynamics underlie the ecosystem responses. In particular, we are interested in the consistency of the soil moisture-ANPP relationship among treatments. Is the relationship steeper under the drought treatment at low soil moisture? Does the relationship saturate under the irrigation treatment at high soil moisture? To answer these questions we fit a generalized linear mixed effects model to test whether the regressions differed among treatments. We also analyzed community composition and the sensitivity of ANPP to drought and irrigation treatments over time, allowing us to place our experimental results within the framework of our scenarios (Fig. 1).

## METHODS

### Study Area

We conducted our precipitation manipulation experiment in a sagebrush steppe community at the USDA-ARS Sheep Experiment Station (USSES) near Dubois, Idaho (44.2°N, 112.1°W), 1,500 m above sea level. The plant community is dominated by the shrub *Artemisia tripartita*, the perennial forb *Balsamorhiza sagittata*, and three perennial bunchgrasses, *Pseudoroegneria spicata*, *Poa secunda*, and *Hesperostipa comata* (see Appendix S1 for rank abundance curves). During the period of our experiment (2011–2016), mean annual precipitation was 250 mm year$^{-1}$ and mean monthly temperature ranged from $-5.2\,°C$ in January to $21.8\,°C$ in July. Between 1926 and 1932, range scientists at the USSES established 26 permanent 1 m$^2$ quadrats to track vegetation change over time. In 2007, we located 14 of the original quadrats in permanenant livestock exclosures, which we used as control plots (i.e., ambient precipitation) in the experiment described below. We used the original plots as our controls because collecting demographic data is time consuming and it was already being collected in these plots for other studies.

In spring 2011, we established 16 new 1 m$^2$ plots located in the largest exclosure at USSES, which also contained six of our control plots. We avoided areas on steep hill slopes, areas with greater than 20% cover of bare rock, and areas with greater than 10% cover of the shrubs *Purshia tridentata* and/or *Amelanchier utahensis*. We established the new plots in pairs and randomly assigned each plot in a pair to receive the "drought" or "irrigation" treatment. Thus, our experiment consisted of $n = 14$ control plots, $n = 8$ drought plots, and $n = 8$ irrigation plots, for a total of 30 plots.

## Precipitation experiment

Drought and irrigation treatments were designed to decrease and increase the amount of ambient precipitation by 50%. To achieve this, we used a system of rain-out shelters and automatic irrigation (*Gherardi & Sala, 2013*). The rain-out shelters consisted of transparent acrylic shingles 1–1.5 m above the ground that covered an area of 2.5 × 2 m. The shingles intercepted approximately 50% of incoming rainfall, which was channeled into 75 liter containers. Captured rainfall was then pumped out of the containers and sprayed on to the adjacent irrigation plot via two suspended sprinklers. Pumping was triggered by float switches once water levels reached about 20 liters. We disconnected the irrigation pumps each October and reconnected them each April. The rain-out shelters remained in place throughout the year.

We monitored soil moisture in four of the drought-irrigation pairs using Decagon Devices (Pullman, Washington) 5TM and EC-5 soil moisture sensors. We installed four sensors around the edges of each 1 × 1 m census plot, two at 5 cm soil depth and two at 25 cm soil depth. We also installed four sensors in areas nearby the four selected plot pairs to measure ambient soil moisture at the same depths. Soil moisture measurements were automatically logged every four hours. We coupled this temporally intensive soil moisture sampling with spatially extensive readings taken with a handheld EC-5 sensor at six points within all 16 plots and associated ambient measurement areas. These snapshot data were collected on 06-06-2012, 04-29-2015, 05-07-2015, 06-09-2015, and 05-10-2016.[1]

Analyzing the response to experimental treatments was complicated by the fact that we did not directly monitor soil moisture in each plot on each day of the experiment. Only a subset of plots were equipped with soil moisture sensors, and within those plots, one or more of the sensors frequently failed to collect data. Therefore, we used a statistical model to estimate average daily soil moisture values for the ambient, drought, and irrigation treatments during the course of the experiment.

We first averaged the observed soil moisture readings for each day ($d$) and plot ($i$), $x_{i,d}$. Experimental plots were located in pairs, with each group ($g$) containing a drought and irrigated plot. In addition, a nearby area outside the drought or irrigated plots was monitored for local ambient soil moisture conditions. Within each group of plots we standardized the irrigation and drought effects on soil moisture relative to the ambient soil moisture conditions. Specifically, we subtracted the ambient daily soil moisture from the soil moisture measured within the drought and irrigation plots within each group and then divided by the standard deviation of daily soil moisture values measured in the ambient conditions $\left( \Delta x_{g,d,\mathrm{trt.}} = (x_{g,d,\mathrm{trt.}} - x_{g,d,\mathrm{ambient}}) / \mathrm{s.d.}(x_{g,d,\mathrm{ambient}}) \right)$ where $\Delta x$ is the

[1] Dates formatted as: mm-dd-yyyy.

standardized treatment effect and $x_{g,d,\text{trt.}}$ is the raw soil moisture measure for plot group $g$ on day $d$ and treatment $trt$. These transformations ensured that the treatment effects in each plot were appropriately scaled by the local ambient conditions within each plot group.

We then modeled these daily deviations ($\Delta x_{g,d,\text{trt.}}$) from ambient conditions using a linear mixed effects model with independent variables for treatment (irrigation or drought), season (winter, spring, summer, fall), rainfall, and all two-way interactions. Using the local daily weather station data, we recorded rainy days as any day with measureable precipitation or the day after such a day and with average temperatures above 3 °C (to exclude days with snowfall). We fit the model using the lme4::lmer() function (*Bates et al., 2015*) in R (*R Core Team, 2016*), with random effects for plot group and date. We weighted observations by the number of unique sensors or spot measurements that were taken in each plot on that day. We then used the model to predict the deviations from ambient conditions produced in the treated plots on each day of the experiment. We added these predicted deviations to the average daily ambient soil moisture to generate predictions for daily soil moisture in all of the treated plots: $\bar{x}_{d,\text{trt.}} = \Delta\bar{x}_{d,\text{trt.}} + \bar{x}_{d,\text{ambient}}$, where $\bar{x}_{d,\text{trt.}}$ is the average predicted daily soil moisture in the treated plots and $\bar{x}_{d,\text{ambient}}$ is the daily ambient soil moisture averaged across all control plots. We could only predict soil moisture in the treated plots on days for which we took at least one ambient soil moisture measurement. This procedure allowed us to predict daily soil moisture conditions for all plots, even on those days when some of our direct treatment measurements were missing due to malfunction of the sensors or data loggers. See *Kleinhesselink (2017)* for more details on our approach to estimating soil moisture.

Following the above procedure, we still lacked soil moisture data for March 2012 (observations did not start until April 2012) and for a string of days in 2013 during which the soil moisture probes failed to take readings. To fill in these gaps, we used a version of the SOILWAT soil moisture model (*Sala, Lauenroth & Parton, 1992*) that has been specifically designed for semiarid shrublands and grasslands (*Bradford, Schlaepfer & Lauenroth, 2014*). The model was parameterized using generic sagebrush steppe parameters and local soil texture, soil bulk density, and precipitation data (*Kleinhesselink, 2017*). We used the locally parameterized SOILWAT model to generate daily soil moisture predictions for the duration of the experiment, but only used SOILWAT predictions where there were gaps in our data (Fig. 2B and Appendix S2). SOILWAT predicted daily soil moisture under ambient conditions similar to our control plots. We applied the same statistical model and procedure described above to estimate soil moisture in drought and irrigation plots based on control plot conditions.

## Data collection

We estimated aboveground net primary productivity (ANPP) using a radiometer to relate ground reflectance to plant biomass (*Byrne et al., 2011*). We recorded ground reflectance at four wavelengths, two associated with red reflectance (626 nm and 652 nm) and two associated with near-infrared reflectance (875 nm and 859 nm). At each plot in each year, we took four readings of ground reflectances at the above wavelengths. We also took readings in 12 (2015), 15 (2012, 2013, 2014), or 16 (2016) calibration plots adjacent to the

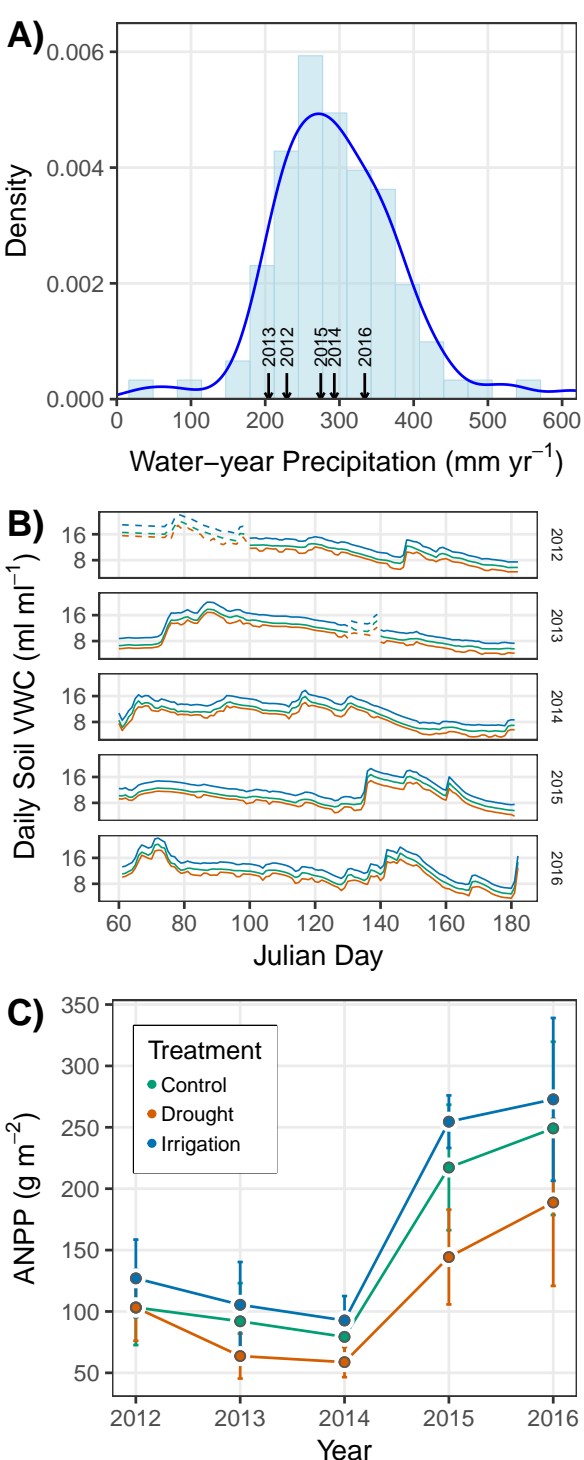

**Figure 2** (A) Probability density of historical water-year (Oct.–Sept.) precipitation from 1927–2016, with the years of the experiment shown by arrows on the *x*-axis. 

**Figure 2 (…continued)**
(B) Statistically estimated (solid lines) and SOILWAT-generated (dashed lines) daily soil volumetric water content (VWC) in each of the three treatments during the course of the experiment. We used the estimates from our statistical model, with gaps filled in by SOILWAT predictions, to calculate cumulative March–June VWC as used in our analysis of treatment effects on ecosystem functional response. (C) Mean (points) ANPP and its standard deviation (error bars) for each year of the experiment. Colors in (B) and (C) identify the treatment, as specified in the legend of (C).

experimental site, in which we harvested all aboveground biomass produced in the current year (we excluded litter and standing dead material), dried it to a constant weight at 60 °C, and weighed it to estimate ANPP. We made radiometer measurements and harvested at peak green biomass each year, typically in late June.

For each plot and year, we averaged the four readings for each wavelength and then calculated a greenness index based on the same bands used to calculate NDVI using the MODIS (Moderate Resolution Imaging Spectroradiometer) and AVHRR (Advanced Very High Resolution Radiometer) bands for NDVI. We regressed the greenness index against the dry biomass weight from the calibration plots to convert the greenness index to ANPP. We fit regressions to a MODIS-based index and an AVHRR-based index for each year and retained the regression with the better fit based on $R^2$ values. We then predicted ANPP using the best regression equation for each year (Appendix S3). Our results do not change when we analyze the soil moisture-NDVI relationship instead of the soil moisture-ANPP relationship (Appendix S5).

Species composition data came from two sources: yearly census maps for each plot made using a pantograph (*Hill, 1920*) and yearly counts of annual species in each plot. From these sources, we determined the density of all annuals and perennial forbs, the basal cover of perennial grasses, and the canopy cover of shrubs. We made a large plot-treatment-year by species matrix, where columns were filled with either cover or density, depending on the measurement made for the particular species. We standardized the values in each column so we could directly compare species quantified with different metrics (density, basal cover, and canopy cover). This puts all abundance values on the same scale, meaning that common and rare species are weighted equally. Assuming that rare species will respond to treatments more than common ones, our approach is biased towards detecting compositional changes. We limited our analysis of community data to observations from the permanent exclosure containing our drought and irrigation treatments (see **Community composition over time**). This exclosure included six of the control plots and all of the treatment plots, for a total of 22 plots. The other eight control plots are in other pastures. Including these plots would add spatial variation in composition, complicating our goal of describing temporal trends in composition.

## Data analysis
### Ecosystem functional response
Our main goal was to test whether the relationship between ANPP and soil moisture differed among the drought, control, and irrigation treatments. Based on our own observations and previous work at our study site (*Blaisdell, 1958*; *Dalgleish et al., 2011*; *Adler, Dalgleish*

*& Ellner, 2012*), we chose to use cumulative volumetric water content from March through June as our metric of soil moisture (hereafter referred to as 'VWC'). We fit a generalized linear mixed effects regression model with log(ANPP) as the response variable and VWC and treatment as fixed effects. Plot and year of treatment were included as random effects to account for non-independence of observations, as described below. We log-transformed ANPP to reduce heteroscedasticity. Both log(ANPP) and VWC were standardized to have mean 0 and unit variance before fitting the model [i.e., $(x_i - \bar{x})/\sigma_x$].

Our model is defined as follows:

$$\mu_i = \boldsymbol{\beta}\mathbf{x}_i + \boldsymbol{\gamma}_{j(i)}\mathbf{z}_i + \eta_t, \tag{1}$$

$$\mathbf{y} \sim \text{Normal}(\boldsymbol{\mu}, \sigma^2), \tag{2}$$

where $\mu_i$ is the deterministic prediction from the regression model for observation $i$, which is associated with plot $j$ and treatment year $t$. $\boldsymbol{\beta}$ is the vector of coefficients for the fixed effects in the design matrix $\mathbf{X}$. Each row of the design matrix represents a single observation ($\mathbf{x}_i$) and is a vector with the following elements: 1 for the intercept, a binary 0 or 1 if the treatment is "drought", a binary 0 or 1 if the treatment is "irrigation", the scaled value of VWC, binary "drought" value times VWC, and binary "irrigation" value times VWC. Thus, our model treats "control" observations as the main treatment and then estimates intercept and slope offsets for the "drought" and "irrigation" treatments. We use our model to test two statistical hypotheses based on the questions in our Introduction:

H1.  The coefficient for drought × VWC is positive and different from zero.

H2.  The coefficient for irrigation × VWC is negative and different from zero.

These hypotheses are based on evidence that precipitation-ANPP relationships often saturate with increasing precipitation (*Hsu, Powell & Adler, 2012*; *Gherardi & Sala, 2015b*).

We include two random effects to account for the fact that observations within plots and years are not independent. Specifically, we include plot-specific offsets ($\boldsymbol{\gamma}$) for the intercept and slope terms and year-specific intercept offsets ($\eta_t$). The covariate vector $\mathbf{z}_i$ for each observation $i$ has two elements: a 1 for the intercept and the scaled value of VWC for that plot and year. The plot-specific coefficients are modeled hierarchically, where plot level coefficients are drawn from a multivariate normal distribution with mean 0 and a variance–covariance structure that allows the intercept and slope terms to be correlated:

$$\boldsymbol{\gamma}_{j(i)} \sim \text{MVN}(0, \Sigma), \tag{3}$$

where $\Sigma$ is the variance–covariance matrix and $j(i)$ reads as "plot $j$ associated with observation $i$". The random year effects ($\boldsymbol{\eta}$) are drawn from a normal prior with mean 0 and standard deviation $\sigma_{\text{year}}$, which was drawn from a half-Cauchy distribution. We used a vague prior distribution for each $\beta$: $\boldsymbol{\beta} \sim \text{Normal}(0, 5)$. A full description of our model and associated R (*R Core Team, 2016*) code is in Appendix S4.

We fit the model using a Bayesian approach and obtained posterior estimates of all unknowns via the No-U-Turn Hamiltonian Monte Carlo sampler in Stan (*Stan Development Team, 2016b*). We used the R package 'rstan' (*Stan Development Team, 2016a*) to link R (*R Core Team, 2016*) to Stan. We obtained samples from the posterior

distribution for all model parameters from four parallel MCMC chains run for 10,000 iterations, saving every 10th sample. Trace plots of all parameters were visually inspected to ensure well-mixed chains and convergence. We also made sure all scale reduction factors ($\hat{R}$ values) were less than 1.1 (*Gelman & Hill, 2009*).

### Sensitivity of ANPP over time

Our data do not allow us to directly test whether the ecosystem functional response to precipitation may have changed over time since the start of the experiment in each of the treatments. This is because we lack sufficient within-year and within-treatment variation of soil moisture (i.e., within a year and treatment each plot shares the same value of VWC). However, we did use a separate analysis to test whether the the sensitivity of ANPP to treatment changed over time.

We define 'sensitivity' as $\frac{\text{ANPP}_{\text{control}} - \text{ANPP}_{\text{treatment}}}{\text{VWC}_{\text{ambient}} - \text{VWC}_{\text{treatment}}}$, following *Wilcox et al. (2017)*. This metric of sensitivity scales the difference between ANPP in the treated and control plots by the change in soil moisture in the treated plots. Because our treated plots are not directly paired with individual control plots, we compare ANPP in each treated plot to the average ANPP of the control plots in each year. After calculating sensitivity for each drought and irrigation plot in each year, we regressed sensitivity against year of treatment using the lm() function in R (*R Core Team, 2016*). This analysis also allows us to link particularly sensitive treatment-years to changes in community composition.

### Community composition over time

We used nonmetric multidimensional scaling (NMDS) based on Bray-Curtis distances to identify temporal changes in community composition among treatments. We first calculated Bray-Curtis distances among all plots for each year of the experiment and then extracted those distances for use in the NMDS. Some values of standardized species' abundances were negative, which is not allowed for calculating Bray-Curtis distances. We simply added '2' to each abundance value to ensure all values were greater than zero. We plotted the first two axes of NMDS scores to see if community composition overlapped, or not, among treatments in each year. We used the vegan::metaMDS() function (*Oksanen et al., 2017*) to calculate Bray-Curtis distances and then to run the NMDS analysis. We used the vegan::adonis() function (*Oksanen et al., 2017*) to perform permutational multivariate analysis of variance to test whether treatment plots formed distinct groupings. To test whether treatment plots were equally dispersed, or not, we used the vegan::betadisper() function (*Oksanen et al., 2017*).

We conducted the above analysis for all species, and then conducted a separate analysis for annual species only (Appendix S1). Annual species have shorter life spans, so conducting a separate analysis allowed us to test whether we might find stronger evidence for community responses to altered precipitation when we focus on short-lived species. Given the dominance of perennial species in our system (Appendix S1), shifts in the annual plant community could be difficult to detect in the analysis of the full community.

*Reproducibility*

All R code and data necessary to reproduce our analysis has been archived on Figshare (https://doi.org/10.6084/m9.figshare.5909998.v1) and released on GitHub (https://github.com/atredennick/usses_water/releases/v0.1). We also include annotated Stan code in our model description in Appendix S4.

## RESULTS

Ambient precipitation and soil moisture were variable over the five years of the study (Figs. 2A–2B). ANPP varied from a minimum of 77.4 g m$^{-2}$ in 2014 to a maximum of 239.3 g m$^{-2}$ in 2016 when averaged across treatments (Fig. 2C). ANPP was slightly higher in irrigation plots (on average 15% higher) and slightly lower in drought plots (on average 25% lower) relative to control plots (Fig. 2C), corresponding to observed and estimated soil volumetric water content (VWC) differences among treatments (Fig. 2B). March–June VWC in drought plots was 12% less than in control plots on average, and March–June VWC in irrigation plots was 19% higher than in control plots on average across the years of the experiment. The differences in soil VWC indicate our treatment infrastructure was successful. ANPP was highly variable across plots within years, as indicated by the large and overlapping standard deviations (Fig. 2C).

Cumulative March–June soil moisture had a weak positive effect on ANPP (Table 1; Fig. 3C). The effect of soil moisture for each treatment is associated with high uncertainty, however, with 95% Bayesian credible intervals that broadly overlap zero (Table 1). Although the parameter estimates for the effect of soil moisture overlap zero, the posterior distributions of the slopes all shrank and shifted to more positive values relative to the prior distributions (Fig. A3-2), which indicates the data did influence parameter estimates. Ecosystem functional response was similar among treatments (Table 1; Fig. 3C), but there is evidence that the slope for the drought treatment is greater than the slope for the control treatment. This evidence comes from interpreting the posterior distributions of the slope offsets for the treatments. From these distributions, we calculate a 42% one-tailed probability that the estimate is less than zero for the irrigation treatment and a 100% one-tailed probability that the estimate is greater than zero for the drought treatment (Fig. 3B).

Sensitivity of ANPP to irrigation was constant over the course of our experiment (Fig. 3D). Sensitivity of ANPP to drought, however, grew over time ($P = 0.0007$; Fig. 3D).

Community composition was similar among treatments. The multidimensional space of community composition overlapped among treatments in all years and was equally dispersed in all years (Table 2; Fig. 4). Community composition was also remarkably stable over time, with no evidence of divergence among treatments (Table 2; Fig. 4). There were also no changes in the dominant species over time in any treatment (Appendix S1). Analyzing annual species on their own produced similar results. There is some evidence that the annual community changed in response to treatment in two years (Table A1-1; Fig. A1-5), but these responses appear to come from very rare species since our analysis weights common and rare species equally (Fig. A1-4). By definition, these rare species

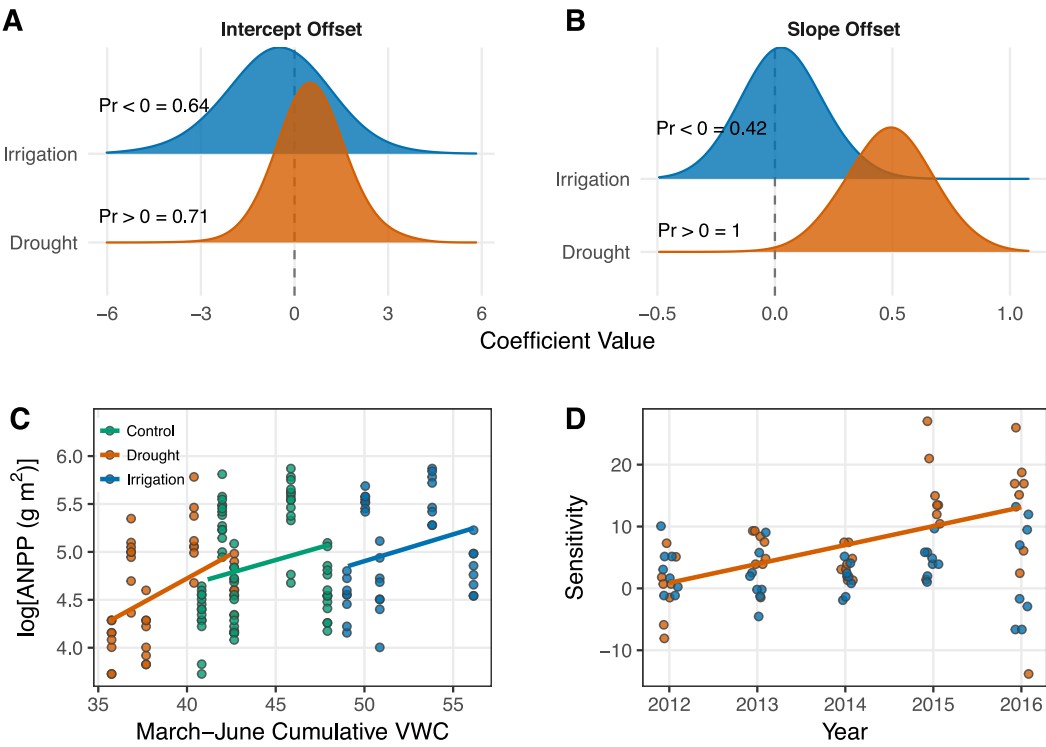

**Figure 3** **Results from the generalized linear mixed effects model (A–C) and the sensitivity analysis (D).** (A–B) Posterior distributions for the effects of drought and irrigation on the intercept (A) and slope (B) of the productivity-soil moisture relationship. Treatment effects show the difference between the coefficients estimated in the treated plots and the control plots. Probabilities ("Pr ⟨/⟩0 =") for each coefficient indicate the one-tailed probability that the coefficient is less than or greater than zero, depending on the median of the intercept offset distributions or our specific hypothesis for the slope offsets. The posterior densities were smoothed for visual clarity by increasing kernel bandwidth by a factor of five. (C) Scatterplot of the data and model estimates shown as solid lines. Model estimates come from treatment level coefficients (colored lines). Note that we show log(ANPP) on the $y$-axis of B; this same plot can be seen on the arithmetic scale in Fig. A4-1. (D) Regression of sensitivity against time for each treatment. Each point represents the sensitivity of ANPP in a plot relative to the mean of the control plots in that year. Only the significant regression for the drought treatment is plotted ($P = 0.0007$).

**Table 1** **Summary statistics from the posterior distributions of coefficients for each treatment ($\beta$ coefficients in Eq. (1)).** The 'Intercept' and 'Slope' summaries reported here for drought and irrigation are from the posterior distributions of the intercept and slope for the control treatment plus the offsets for each treatment. Posterior distributions of the offsets are in Fig. 3A.

| Coefficient | Treatment | Posterior mean | Posterior median | Lower 95% BCI | Upper 95% BCI |
|---|---|---|---|---|---|
| Intercept | Control | 0.14 | 0.14 | −1.24 | 1.50 |
| Intercept | Drought | 0.67 | 0.65 | −2.01 | 3.36 |
| Intercept | Irrigation | −0.39 | −0.34 | −3.51 | 2.48 |
| Slope | Control | 0.53 | 0.52 | −1.44 | 2.66 |
| Slope | Drought | 1.02 | 1.00 | −1.04 | 3.25 |
| Slope | Irrigation | 0.56 | 0.55 | −1.43 | 2.71 |

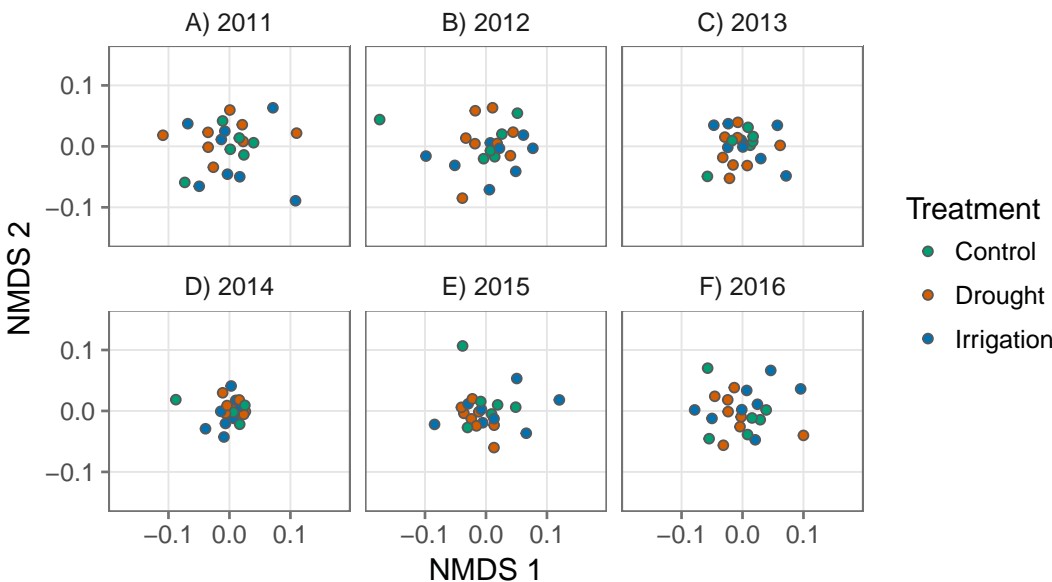

**Figure 4** **Nonmetric multidimensional scaling scores representing plant communities in each plot for each year, colored by treatment.** Each point represents a plot and the axes represent species composition.

**Table 2** **Results from statistical tests for clustering and dispersion of community composition among precipitation treatments.** 'Adonis' tests whether treatments form unique clusters in multidimensional space; 'betadisper' tests whether treatments have similar dispersion. For both tests, *P* values greater than 0.05 indicate there is no support for the hypothesis that the treatments differ.

| Year | Test | *n* | *d.f.* | *F* | *P* |
|------|------|-----|--------|-----|-----|
| 2011 | adonis | 22 | 2 | 1.04 | 0.43 |
| 2011 | betadisper | 22 | 2 | 2.22 | 0.14 |
| 2012 | adonis | 22 | 2 | 1.09 | 0.33 |
| 2012 | betadisper | 22 | 2 | 0.21 | 0.81 |
| 2013 | adonis | 22 | 2 | 1.26 | 0.13 |
| 2013 | betadisper | 22 | 2 | 0.43 | 0.66 |
| 2014 | adonis | 22 | 2 | 0.96 | 0.57 |
| 2014 | betadisper | 22 | 2 | 0.36 | 0.70 |
| 2015 | adonis | 22 | 2 | 1.08 | 0.34 |
| 2015 | betadisper | 22 | 2 | 1.96 | 0.17 |
| 2016 | adonis | 22 | 2 | 1.12 | 0.28 |
| 2016 | betadisper | 22 | 2 | 0.27 | 0.76 |

contribute little to ANPP and thus any compositional changes have little influence on ecosystem functional response.

# DISCUSSION

Ecosystem response to a new precipitation regime depends on the physiological responses of constituent species and the rate at which community composition shifts to favor species

better able to take advantage of, or cope with, new resource levels (*Smith, Knapp & Collins, 2009*). Previous work has shown that community compositional shifts can be either rapid, on the order of years (*Hoover, Knapp & Smith, 2014*), or slow, on the order of decades (*Knapp, Briggs & Smith, 2012*; *Wilcox et al., 2016*). A lingering question is how the time scales of ecosystem response and community change vary among ecosystems. Precipitation manipulation experiments can help answer this question, especially if they push water availability outside the historical range of variability for long periods.

We found that ecosystem functional response under chronic drought was different from the control treatment, but community composition remained unchanged (Fig. 3B, Fig. 4), representing the scenario in Fig. 1B. The increase in the slope of the VWC × productivity relationship in the drought plots indicates increased sensitivity to water availability under chronic drought. A strict interpretation of this result implies that if average soil moisture were pushed consistently lower than currently observed ambient conditions, there would be a stronger relationship between precipitation and productivity in this system.

However, we do not want to over-interpret the significance of the slope offsets given that the overall slopes of the VWC × productivity relationship, not just the offsets, were similar among treatments (Table 1). We therefore conclude that ecosystem functional response is consistent (similar values) and weak (all broadly overlapping zero) across all precipitation treatments.

The similarity of ecosystem functional response (Table 1; Fig. 3C) and community composition (Fig. 4) among treatments is surprising because grasslands generally, and sagebrush steppe specifically, are considered water-limited systems. For example, *Huxman et al. (2004)* and *Knapp et al. (2015)* showed that semi-arid sites are more sensitive to drought than mesic sites, and *Wilcox et al. (2017)* found that semi-arid sites are particularly sensitive to irrigation treatments. Based on these findings, we expected ecosystem functional response, community composition, or both to change under our treatments. Why did our treatments fail to induce ecosystem or community responses? We can think of three reasons. Two are limitations of our study, and one invokes the life history traits of the species at our study site.

First, our precipitation manipulations may not have been large enough to induce a response. A 50% decrease in any given year may not be abnormal given our site's historical range of variability (*Knapp et al., 2017*). We cannot definitively rule out this possibility, but we have reason to believe our treatments *should* have been large enough. Using the methods described by *Lemoine et al. (2016)*, we calculated the percent reduction and increase of water-year precipitation necessary to reach the 1% and 99% extremes of the historical precipitation regime at our site (Fig. A5-1). The 1% quantile of water-year precipitation at our site is 78 mm, a 26% reduction from the mean, and the 99% quantile is 545 mm, a 84% increase from mean growing season precipitation (Appendixn S6). Thus, our drought treatment represented extreme precipitation amounts, and that is the treatment where we observed a small effect on the slope between soil moisture and ANPP (Fig. 3B). The irrigation treatment may not have been extreme enough to induce a response.

Second, ANPP at our site is likely influenced by additional factors, not only the cumulative March–June soil moisture covariate we included in our statistical model.

For example, *La Pierre et al. (2016)* found that site-scale ANPP is better predicted by nutrient availability than precipitation. Moreover, temperature can impact ANPP directly (*Epstein, Lauenroth & Burke, 1997*) and by exacerbating the effects of soil moisture (*De Boeck et al., 2011*). Measurements of soil moisture likely contain a signal of temperature, through its impact on evaporation and infiltration, but the measurements will not capture the direct effect of temperature on metabolic and physiological processes. We also did not redistribute snow across our treatments in the winter, and snow melt may spur early spring growth. Failure to account for potentially important covariates could explain the within-year spread of ANPP (Figs. 2C, 3C) and the resulting uncertain relationship we observed between soil moisture and ANPP across all treatments (Table 1).

Third, the life history traits of the dominant species, which largely determine ecosystem function (*Smith & Knapp, 2003*), may explain the weak and uncertain effect of soil moisture on ANPP (Table 1, Fig. 3). Species that live in variable environments, such as cold deserts, must have strategies to ensure long-term success as conditions vary. One strategy is bet hedging, where species forego short-term gains to reduce the variance of long-term success (*Seger, 1987*). In other words, species follow the same conservative strategy every year, designed to minimize losses during unfavorable periods. The dry and variable environment of the sagebrush steppe has likely selected for bet hedging species that can maintain function at low water availability and have weak responses to high water availability. In so doing, the dominant species in our ecosystem avoid "boom and bust" cycles, which may explain the weak effect of soil moisture on ANPP (i.e., the Bayesian credible intervals for the slopes overlapping zero; Table 1).

Another strategy to deal with variable environmental conditions is avoidance, which would also result in a consistent ecosystem functional response between drought and control treatments. For example, many of the perennial grasses in our focal ecosystem avoid drought stress by growing early in the growing season (*Blaisdell, 1958*; A.R. Kleinhesselink, pers. obs.). Furthermore, the dominant shrub in our focal ecosystem, *Artemisia tripartita*, has access to water deep in the soil profile thanks to a deep root system (*Kulmatiski et al., 2017*). The dominance of our site by the shrub *A. tripartita* (Appendix S1) may explain why our results do not conform to broader patterns of grassland sensitivity to precipitation manipulations (e.g., *Huxman et al., 2004*; *Knapp et al., 2015*; *Wilcox et al., 2017*).

We found that ANPP became more sensitive to the drought treatment over time (Fig. 3D). We intereptet this increase in sensitivity as the effect of cumulative impacts of the drought on dominant species. Plants may not have shown a large response in terms of ANPP in the first years of the experiment if they could grow from stored carbohydrate reserves or if they abstained from flowering and reproduction. As the drought progressed, these same plants may have have started to shrink or die. Given the long-lived perennial species at this site, this increase in sensitivity may indicate a larger future change in community composition and ecosystem functional response.

## CONCLUSIONS

Our results suggest that the species in our focal plant community are insensitive to to changes in precipitation regime, at least over the five years of our experiment. Such

insensitivity can buffer species against precipitation variability in this semi-arid ecosystem, making them successful in the long run. Longer, chronic precipitation alteration might reveal plant community shifts that we did not observe (e.g., *Wilcox et al., 2016*), and the increased sensitivity of ANPP to drought in the final years of our experiment may portend such a shift. Likewise, a long-term increase in water availability could allow species that do not bet hedge to gain prominence and dominate the ecosystem functional response. The length of the perturbation may be especially relevant in our focal ecosystem where the perennial species are long-lived, meaning compositional turnover may take many more years than we report on here.

## ACKNOWLEDGEMENTS

We thank the many summer research technicians who collected the data reported in this paper and the USDA-ARS Sheep Experiment Station (Dubois, ID) for facilitating work on their property. We also thank Susan Durham for clarifying our thinking on the statistical analyses and Kevin Wilcox for helpful discussions on analyzing community composition data. John Bradford and Caitlin Andrews generously provided SOILWAT model output. Two anonymous reviewers and Elsie Denton provided thoughtful reviews that improved our paper.

### Funding

This research was supported by the Utah Agricultural Experiment Station, Utah State University, and approved as journal paper number 9035. The research was also supported by the National Science Foundation, through a Postdoctoral Research Fellowship in Biology and Mathematics to Andrew T. Tredennick (DBI-1400370), a Graduate Research Fellowship to Andrew R. Kleinhesselink, and grants DEB-1353078 and DEB-1054040 to Peter B. Adler. The funders had no role in study design, data collection and analysis, decision to publish, or preparation of the manuscript.

### Grant Disclosures

The following grant information was disclosed by the authors:
Utah Agricultural Experiment Station.
Utah State University.
National Science Foundation: DBI-1400370, DEB-1353078, DEB-1054040.

### Competing Interests

Andrew T. Tredennick is an Academic Editor for PeerJ.

### Author Contributions

- Andrew T. Tredennick performed the experiments, analyzed the data, contributed reagents/materials/analysis tools, prepared figures and/or tables, authored or reviewed drafts of the paper.

- Andrew R. Kleinhesselink conceived and designed the experiments, performed the experiments, analyzed the data, contributed reagents/materials/analysis tools, authored or reviewed drafts of the paper.
- J. Bret Taylor conceived and designed the experiments, contributed reagents/materials/-analysis tools, authored or reviewed drafts of the paper.
- Peter B. Adler conceived and designed the experiments, performed the experiments, analyzed the data, authored or reviewed drafts of the paper.

## Data Availability

Tredennick, Andrew; Kleinhesselink, Andrew; Taylor, J. Bret; B. Adler, Peter (2018): Data and code from: Ecosystem functional response across precipitation extremes in a sagebrush steppe. figshare. https://doi.org/10.6084/m9.figshare.5909998.v1.

## Supplemental Information

Supplemental information for this article can be found online at http://dx.doi.org/10.7717/peerj.4485#supplemental-information.

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
