# Peer review of "Ecosystem functional response across precipitation extremes in a sagebrush steppe"

_PeerJ, doi:10.7717/peerj.4485_

## Round 0.1 · original submission · Major Revisions

Please pay close attention to the comments from Reviewer 1 regarding the validity of the findings. Once you have responded to these comments in a point-by-point cover letter and using tracked changes in the manuscript document, I will likely send your manuscript out for re-review.

Reviewer 1 ·

Basic reporting

The basic reporting is quite good. The manuscript is well-written and easy to follow. The statistics and pre-treatment analyses were, despite being complicated, easy to understand. No comments or concerns here.

Experimental design

The study was well executed. I particularly like the use of the Sala experimental design of automated drought+irrigation treatments. I think this is a powerful design that enables the authors to speak more thoroughly about the role of water limitation in this system (as opposed to a drought-only experiment). The author also did an extraordinary amount of work on the statistal analyses and to resolve issues (VWC measurements/ANPP calculations) in a quantitatively rigorous manner.

Validity of the findings

Their results indicate that the sagebrush steppe is actually quite robust to interannual variability in precipitation. This comes as a surprise, as the literature suggests that arid and semi-arid ecosystems (into which the sagebrush steppe easily fits) are quite sensitive to changes in precipitation. Huxman et al. (2004) and Knapp et al. (2015) both showed that drought impacts semi-arid sites more intensely than mesic sites, and Wilcox et al. (2017, in GCB) showed that most systems are more sensitive to irrigation than drought, and this is especially true for semi-arid sites. So far as I can tell, this study doesn't fit neatly into either line of thought (since it was relativel insensitive to drought, and irrigation effects were actually weaker than drought - see Wilcox et al.). It would be beneficial for the authors to put their study into this larger context and perhaps discuss why they think their site doesn't conform to these broader patterns (i.e. shrub-dominated, etc.).

I also have a couple of concerns that potentially influence the results that the authors need to address prior to publication.

1. I appreciate the authors effort in validating and calibrating their NDVI measurements to ANPP, but I am quite concerned about the accuracy of these calibrations. The R2 is quite low for those calibration regressions (0.2 - 0.7), so I wonder how accurately biomass in each plot was represented by these ANPP measures. This is actually quite a large source of error, and I wonder if it isn't responsible for the lack of significance and weak responses observed here. I have two thoughts: would it be better to use raw NDVI measures rather than ANPP conversions? could the authors imbed the ANPP-NDVI calibration into their regressions (see Kiona Ogle's papers on multiple datasets in Bayesian analyses)? This would allow full uncertainty around the estimated ANPP based on NDVI to propogate into the VWC analyses.

2. I didn't see mention of the priors on B, but I might've missed it. See Lemoine et al. (2016) - Ecology for a discussion of this issue. (very minor issue, I doubt the priors make too much difference here).

3. My biggest concern is that the statistical tests don't seem to test the HRF because they treat year/time as a "nuisance" variable that adds noise to the observations. But the HRF is explicitly about time and how the response of ecosystem function changes over time. So incoporating time as an explicit part of the analysis would be necessary. The easiest way would be to regress ANPP against VWC and allow the slope to vary by year to see if the slopes follow the pattern dictated by HRF (or could use a hierarchical gaussian process/LOWESS model or other non-parametric model to allow for non-lienarity along the VWC gradient - drought - control - irrigation). Perhaps another approach would be to calculate a 'sensitivity' for each replicate (or treatment) within a Bayesian model, then look at how sensitivity to drought/irrigation changes through time (in the same model, i.e. plot-level sensitivity nested within year, with fixed effects to contrast sensitivity to drought/irrigation, a la Wilcox et al. 2017). That is an explicit test of HRF, where the sensitivites could show any number of temporl patterns outlined by Smith.

Reviewer 2 ·

Basic reporting

• Well-presented manuscript. I appreciated the additional details provided in the appendices, particularly appendix 2.
• A more general ecology audience may benefit if the introduction would start with an additional first paragraph to provide a framing of the research in a larger picture and to explain why the ANPP--soil moisture relationship is so important.

Experimental design

• I don’t understand the rationale for why the analysis is presented in terms of ANPP responses whereas NDVI was measured. The ANPP values represent a linear transformation of the NDVI measurements, but with a mediocre to low R2 (A1-2), e.g., R2 for year 2015 was 0.21 -- an information that is somewhat buried in the supplementary files. This less than ideal relationship is not discussed, e.g., as potential fourth source for not finding responses (lines 285-288). Why not analyzing NDVI--soil moisture relationships instead/in addition to ANPP--soil moisture relationships?

Validity of the findings

• The interpretation/discussion is coherent with the results. Even though the study found no responses, I agree that this is a reasonable outcome for a shrub-dominated, semi-arid ecosystem.
• The introduction presents the idea that there are four basic types of how ANPP responds to increases/decreases in soil moisture and that the time it takes to express a response will differ among different communities (lines 87-90). Even though the discussion section suggests that species life history traits (‘bet hedging’, lines 310-319) are important to understand the results of no treatment responses, I feel that it would be as important to discuss how the lifespan of species in the community in relation to the experimental duration of the treatments promotes/hinders the observation of responses.
o Several of the dominant species (e.g., threetip sagebrush and bluebunch wheatgrass) have a lifespan that is much longer than the 5-year experiment; while other species in the community have shorter reproductive cycles. It seems that even with a 5-year experiment the longer-lived species of the experimental community may not have had sufficient time to (fully) respond and make it possible to observe measurable changes in the community composition/response.
o I would find it interesting if the plant community analysis is additionally carried out separately for short-lived vs long-lived species (Fig. 4) -- based on the expectations that short-lived species have more ‘opportunities’ to respond/turnover than long-lived species.

Additional comments

• Fig. 3: Misspelling in ‘increasing kernal bandwidth’ -- this should be ‘kernel’
• Fig 4: Expand caption: explain that dots represents plots and dimensions represent species.

·

Basic reporting

Clear and unambiguous, professional English used throughout.
The article must be written in English using clear and unambiguous text and must conform to professional standards of courtesy and expression.
Literature references, sufficient field background/context provided.
The article should include sufficient introduction and background to demonstrate how the work fits into the broader field of knowledge. Relevant prior literature should be appropriately referenced.

- English is high quality with very few typos, though a few locations in text could use rephrasing to increase clarity. Some restructuring of the introduction is suggested to provide a more logical and engaging presentation of the topic. Whenever I had a question about a topic the authors were referring to, or a method they were using, their citations were sufficient for me to conduct further investigation. More discussion of prior work into the role of dominant species in determining ecosystem function may be warranted.

Professional article structure, figs, tables. Raw data shared.
The structure of the article should conform to an acceptable format of ‘standard sections’ (see our Instructions for Authors for our suggested format). Significant departures in structure should be made only if they significantly improve clarity or conform to a discipline-specific custom.

- There are no significant departures from standard structure.

Figures should be relevant to the content of the article, of sufficient resolution, and appropriately described and labeled.

- Overall figures are well designed and easy to read, but treatment colors don’t translate well into greyscale. More contrasting colors (in terms of light and dark) are suggested. Minor things about a couple of table and figure headings could be changed to improve clarity. See notes to authors.

All appropriate raw data has been made available in accordance with our Data Sharing policy.

- Data is easy to find and download.

Self-contained with relevant results to hypotheses.
The submission should be ‘self-contained,’ should represent an appropriate ‘unit of publication’, and should include all results relevant to the hypothesis.
Coherent bodies of work should not be inappropriately subdivided merely to increase publication count.

- Product is a self-contained research unit that doesn’t leave the reader wondering where the rest of the results are.

Experimental design

Original primary research within Aims and Scope of the journal.

- Primary research in Environmental science.

Research question well defined, relevant & meaningful. It is stated how research fills an identified knowledge gap.

- The research goal is to investigate how the sagebrush steppe ecosystem may respond to long-term climate extremes outside of historic norms. Since most predictive work is currently done using long-term correlations between ecosystem function and climate, but the climate is changing this is a pressing concern. The inherent variability of ecological responses across ecosystems means that each ecosystem needs assessment.

Rigorous investigation performed to a high technical & ethical standard.

- Design of the experiment looks very much like it was completely specified ahead of time and largely adhered to throughout. Where adjustments to the initial design were made, (probably in the soil moisture modeling) the procedure by which this was done was completely laid out. Though this is one of the areas I thought could use some rephrasing to increase clarity. The modeling work is excellent and looks at many different aspects of the question.

Methods described with sufficient detail & information to replicate.

- Generally, the methods are quite detailed, including the description of the statistical analysis so reproduction should be possible. There were a few sections I thought could use additional clarification and these have been addressed in the comments to the authors.

Validity of the findings

Impact and novelty not assessed. Negative/inconclusive results accepted. Meaningful replication encouraged where rationale & benefit to literature is clearly stated.

- Many of the findings of this paper were negative, i.e. no change in ecosystem function or composition was observed. The authors draw the appropriate conservative conclusions of no effect. They do however discuss the possible implications of the positive and negative lean in the data, which might make for interesting grounds for future research.

Data is robust, statistically sound, & controlled.

- Sample size is not large, though neither is it tiny. It was analyzed in such a way that information could still be learned despite the smaller sample size. Appropriate controls were included, though I would like a bit more information on why they used the controls they did, see comments to authors.

Conclusion are well stated, linked to original research question & limited to supporting results.

- Conclusions are clear and do not reach beyond the scope of the data.

Speculation is welcome, but should be identified as such.

- Authors speculated at length regarding why negative results were found. Concluding that it was likely a property of the ecosystem and probably not an artifact of the way the experiment was carried out, though they were careful to state that this possibility could not be ruled out.

Additional comments

This is a good article that yields insight into how experimental work can allow us to test whether historic precipitation/biomass relationships will told up to potential future changes in the precipitation regime. Overall, this is a very strong study and paper; it will make an excellent contribution to the scientific literature. However, there are some small issues that should be resolved first. There are some organizational issues in the Introduction; particularly, it seems the motivation for the study is buried deep within the text instead of in the first paragraph. Revising this section would make the paper a much more compelling read. Additionally, there were a number of places in the methods where I felt that additional clarity was needed so that others could replicate their study. Below I give line-by-line suggestions on these points and other minor editing suggestions. This reviewer feels that this paper should be accepted with minor revisions.

General comments to authors

Lines 31-32: sentence beginning with “However, ANPP response…” seems fairly vague and weakly worded, consider revising to increase clarity

Introduction:
While the abstract opens with a strong justification tying the paper into shifting precipitation patterns as a result of climate change and the need to develop a way to predict how communities might change with long-term exposure, the introduction opens only with a statement saying that there is a relationship between soil moisture and ANPP. This seems incredibly weak and uncompelling. The authors should consider restructuring their introduction so that the justification for their study is at the beginning. They have several other paragraphs in the paper that could do nicely as an opening paragraph. The opening paragraph of the discussion Lines 262-269 would make for an excellent opening. Or, with a little retooling the second paragraph of the introduction (lines 53-64) would make for a better statement of research motivation than the current opening.
Line 46 “outside” is repeated twice
Lines 71-72. While this is an excellent and clear definition of ‘ecosystem functional response’, I think it might be prudent at some point to explain why the authors are using soil moisture as an explanatory variable for ANPP instead of soil moisture. I agree that soil moisture is generally a better predictor of ANPP than direct precipitation measures, but most of the articles the authors cite refer to
ANPP-precipitation relationships, so taking a moment to explain why they are using soil moisture would be warranted.
Lines 88-90: another good motivating factor for why this research was done that could be used earlier in the introduction.
Final paragraph on introduction. I really like the way the authors are testing multiple hypothesis based on theory.

Methods:
110-113: Could the authors please explain why these long-term monitoring plots make better controls than just say a generic 1 m2 plot within the exclosures next to the experimental plots. Certainly the long-term record on community shifts in these plots are likely interesting in their own right, but it doesn’t seem like they add much to the current study.
Lines 126-127: It does concern me that the authors didn’t account for snow/winter precipitation on their irrigation plots. It appears as though a large percentage of the yearly precipitation in Dubois, ID falls over winter. Additionally, since their drought shelters were left up over winter they may actually have attracted drifting and captured more snow on the plots. To their credit, the authors did address this issue to some extent in the Discussion, lines 305-306.
Lines 132-135: Wasn’t initially clear that these were manually collected data instead of another set of sensors. Maybe mention that they were collected by handheld probe at the beginning of the description instead of at the very end.
Lines 142-147: Description of standardization procedure is confusing. I had to read it three times before the procedure was clear to me. Consider rewording for clarity.
Lines 148-158: I’m a bit confused on which plots this modeling was conducted for. Was it done only on those plots that had permanent sensors in place? Or when possible was it also extrapolated to include those plots that only had point in time measurements. Please clarify.
Lines 168-174: I like the idea of using the two separate estimates of greenness and then using the one that predicts best every year. Very thorough.
Line 188: later, in figure 2, the authors refer to mean volumetric water content as “VWC” also. It would be best to use a different abbreviation for the two separate uses of volumetric water content: mean and cumulative.

Results:
Lines 241-244: Even though figures are referred to here, it might be nice to have some measure of the effect size in text, perhaps percentages, just to give a numerical estimation of differences, or lack there of, between plots in ANPP and soil moisture.
Lines 257-260: the stability of the community over the study was impressive particularly given a method sensitive to rare species was used. It would be nice to see a bit more description of what the community looked like. How much of the community was the dominants, vs. rarer species?

Discussion:
Lines 279-281: I like that the authors were very conservative in their eventual conclusions despite their discussion of the potential meanings of the parameter estimates when these simply leaned positively or negatively and all overlapped zero.
Lines 289-299: strong discussion and support for the sizes of the treatment applied. Good work.
Lines 320-325: seems like authors are indicating that ecosystem response was driven by the response of the dominant species. Might be good to bring in some other work that has found similar results and discuss it a bit, for example: “Drivers of Variation in Aboveground Net Primary Productivity and Plant Community Composition Differ Across a Broad Precipitation Gradient” (La Pierre et al., 2016).

Figures and tables:
Table 1: These are the beta coefficents from the model correct? Might be good to elaborate a bit more in the table description.
Figure 2: Here is the use of VWC as “mean soil volumetric water content” I referred to earlier. Like I said it would probably be best to have different abbreviations for different metrics of soil moisture. Also interesting, though not the point of this paper, after three consecutive dry years with progressively decreasing ANPP it looks like the ecosystem was primed for recovery as soon as an average precipitation year hit in 2015. No evidence of drought legacy.
Figures general: differences between colors used to indicate treatments are not at all distinguishable if viewed in greyscale. Authors should consider using a color scheme with more contrast.

---

## Round 0.2 · accepted · Accept

Thank you for your comprehensive responses to the reviewer comments. The reviewers and I are very happy with how you dealt with the comments and feel that the manuscript is now ready for publication (note the minor comment below from Reviewer 2).

Reviewer 1 ·

Basic reporting

As before, the reporting and writing are very clear

Experimental design

As before, the experimental design is good. I doubt it changed during revision.

Validity of the findings

The authors have done an excellent job of addressing my concerns. The sensitivity analysis adds a new aspect to the study, and the authors did a great job discussing why their results do not conform to the expected patterns from other studies.

Additional comments

Thanks for the hard work! Nice job on the revisions. -Nate

Reviewer 2 ·

Basic reporting

A thoughtfully revised and improved manuscript; addressed all comments of reviewers with care and detail

Experimental design

Complete

Validity of the findings

Complete

Additional comments

- line 125: "n = 8 irrigation plots, and n = 8 irrigation plots" í one of two sets of n = 8 should be drought plots?